# In Vitro Handling Characteristics of a Particulate Bone Substitute for Ridge Preservation Procedures

**DOI:** 10.3390/ma17020313

**Published:** 2024-01-08

**Authors:** Samira Dahl, Virgilia Klär-Quarz, Annika Schulz, Matthias Karl, Tanja Grobecker-Karl

**Affiliations:** Department of Prosthodontics, Saarland University, 66421 Homburg, Saar, Germany; samira_dahl@web.de (S.D.); virgilia.klaer@uks.eu (V.K.-Q.); annika.schulz2@uks.eu (A.S.); matthias.karl@uks.eu (M.K.)

**Keywords:** ridge preservation, bone substitute materials, alveolar ridge augmentation

## Abstract

While particulate bone substitute materials are applied in a variety of augmentation procedures, standardized defects are being used for preclinical testing. This in vitro study evaluated the density and homogeneity of a particulate bone substitute in ridge preservation procedures. Premolars and molars were extracted in ten semimandibles of minipig cadavers. Light body impression material was used for determining the volume of the extraction sites followed by augmentation with particulate material, thereby weighing the graft material needed. Microradiographs and histologic sections were obtained for evaluating the homogeneity and density of the augmentation material. Statistical analyses were based on Shapiro–Wilk tests, Spearman’s rho and one sample Wilcoxon test followed by Bonferroni–Holm correction for multiple testing (α = 0.05). Based on 103 single alveoli evaluated, the mean volume determined was 0.120 cm^3^ requiring a mean amount of graft material of 0.155 g. With only three exceptions, all parameters (volume, mass of augmentation material, density and homogeneity) correlated significantly (*p* < 0.020). The apical parts of the alveoli showed reduced density as compared to the middle parts (*p* < 0.001) and the homogeneity of the augmentation material was also lower as compared to the middle (*p* < 0.001) and cervical parts (*p* </= 0.040). The packing of augmentation material is critical when non-standardized defects are treated.

## 1. Introduction

Prosthetic-driven implant placement frequently requires bone augmentation [1,2,3] or at least the prevention of bone loss following tooth extraction [4,5,6,7]. Similarly, bone augmentation has been described for implants immediately placed in extraction sockets for filling up empty spaces between implants and socket walls [8]. Such situations are comparable to crater-like peri-implantitis lesions [9], which are not only difficult to assess during preoperative imaging [10,11] but also require regenerative treatment following debridement and disinfection [12,13].

While the choice of augmentation technique and biomaterial depends on the defect morphology [2,14], particulate bone substitute materials [1,15] play an important role in implant-related surgeries [14]. Heavily affecting the regenerative potential [16], the shape of the granules [17], pore size within the granules, as well as particle size [18] and interconnectivity of pores have received lots of attention [19,20] in order to optimize the vascularization of the graft [21] and subsequent mechanical bone quality [22,23].

It has been argued based on a clinical study that the complete fill of random-shaped bony defects cannot be achieved predictably [9]. As one of the few options for intervention during augmentation, but at the same time a critical factor, the level of compaction of augmentation material cannot be standardized and depends on the clinical experience of the surgeon [24,25]. In a recent animal study, it was shown that compressive forces in the range of 200 g would facilitate the penetration of particulate graft materials into apical regions of sockets and defects and hence optimize bone formation [24,26].

This basic problem of adapting hard bone filling material to a specific surgical site is already well known from the non-dental field [27]. Using calcium phosphate-based composites in the form of injectable bone substitutes may constitute a convenient alternative, thereby avoiding the problem of incomplete or non-homogenous socket filling. Calcium phosphates have been investigated due to their similarity to the mineral component of bone [27] and various attempts have already been made in order to optimize the handling properties of the material [28]. An interesting approach was described and tested by Klijn et al. [29] adding NaHCO_3_, Na_2_HPO_4_ and NaH_2_PO_4_ in specific concentrations and in a strict order to calcium phosphate, which led to the production of CO_2_ from NaHCO_3_ [29]. The rationale for this approach was to introduce adequate pores and interconnectivity into the bone substitute.

From a scientific and regulatory point of view, bone augmentation materials are often tested in standardized defects [22,30] in order to allow for a quantitative analysis of healing processes [31,32], which does not reflect reality where, e.g., extraction sockets [26] with variable size and shape are present [5,30], which may affect the regenerative potential.

The primary goal of this animal cadaver study was to evaluate intraalveolar voids being present following augmentation with one specific particulate bone substitute material by determining homogeneity and density. From a methodologic point of view, a secondary endpoint was to compare microradiographic and histologic analysis.

## 2. Materials and Methods

Ten semi mandibles of adult Aachen minipig cadavers [25,30] were obtained (Heinrichs Tierzucht, Heinsberg, Germany) and the premolars P3 and P4 as well as the first molar M1 were carefully extracted using standard instruments. The premolars had two roots each (mesial and distal) while the molars had four roots (mesiolingual, distolingual, mesiobuccal and distobuccal). The complete extraction of all roots was verified using periapical radiographs (Heliodent, Dentsply Sirona, York, PA, USA/VistaScan image plates, Dürr Dental, Bietigheim-Bissingen, Germany).

Light body silicone impression material (Silasoft, Detax, Ettlingen, Germany; density: 1.17 g/cm3) was subsequently injected into the alveoli and allowed to fully set. After removal, the silicone impressions were weighed for determining the accessible volume. Mean values based on five impressions were noted for statistical analyses.

Particulate bone augmentation material (Creos, Nobel Biocare, Gothenburg, Sweden) was then used for filling the single alveoli, thereby measuring the amount of material needed until the ridge crest is reached. The alveoli were subsequently covered with wax paper mimicking coverage with a membrane in order to avoid the loss of granules during subsequent processing.

All bone specimens were fixed in 10% neutral buffered formalin for 48 h and reduced to small blocks using a diamond band saw (EXAKT 300, EXAKT Advanced Technologies GmbH, Norderstedt, Germany). The specimens were then dehydrated in alcohol solutions of increasing concentrations, clarified in xylene and embedded in polymethylmethacrylate (Technovit 9100, Heraeus Kulzer, Hanau, Germany). One vertical cross-section was obtained per specimen using a cutting and grinding technique [33]. With the sections reduced to a thickness of 120 µm, microradiographs (Faxitron X-ray, Lincolnshire, IL, USA; 14 kV, 0.3 mA, 2.5 min; VistaScan image plates) were made. Following a further reduction of the sections to a thickness of 50–80 µm and staining with toluidine blue O solution after preprocessing in 10% H_2_O_2_, the samples were inspected using a microscope (LEICA DM4B, LEICA Mikrosysteme Vertrieb GmbH, Wetzlar, Germany) equipped with a color image analyzing system (LEICA Application Suite, LEICA Phase Expert, LEICA Mikrosysteme Vertrieb GmbH). A representative image of each alveolus was taken depicting its complete outline. Both microradiographs and histologic sections were then evaluated by three independent examiners with the goal of rating the homogeneity (1 = no; 2 = partly; 3 = yes) and density (1 = low; 2 = medium; 3 = high) of the augmentation material in the apical, middle and cervical third of each alveolus.

Statistical analyses were based on Shapiro–Wilk tests on the normal distribution of measurement values and ratings followed by calculating Spearman’s rank correlation coefficients between variables and one-sample Wilcoxon tests (Mann–Whitney tests) for comparisons. Given that ratings in different regions of the alveoli [24] could not be considered as being independent, rating differences (middle-apical; middle-cervical; and apical-cervical) were tested with respect to differing from zero. Correction for multiple testing was performed according to the Bonferroni–Holm method and the level of significance was set at α = 0.05.

## 3. Results

A total of 103 single alveoli were evaluated and the mean values and standard deviations for the volume of the alveoli (Figure 1) and mass of augmentation material required (Figure 2) as well as for ratings of density and homogeneity (Figure 3) are given in Table 1.

Shapiro–Wilk tests showed significant values (*p* < 0.040) for all parameters indicating a non-normal distribution requiring a non-parametric correlation test (Spearman’s rho). While correlation coefficients varied widely (Table 2), significant correlations were found after Bonferroni correction for all combinations of parameters with the following exceptions: Volume/Homogeneity—histology (*p* = 0.060); Density—microradiograph/Homogeneity—histology (*p* = 0.060); and Homogeneity—microradiograph/Homogeneity—histology (*p* = 0.060).

Separating the cervical, middle and apical thirds of alveoli, mean values for density and homogeneity were calculated and differences between these regions were expressed as *p*-values (Table 3). In the apical regions (Figure 4a), significantly lower density of the augmentation material was reached as compared to the middle part of the alveoli (microradiograph *p* < 0.001; histology *p* < 0.001). In addition, the middle section showed greater density as compared to the cervical section in histology (*p* = 0.020) but not in microradiographs (*p* = 0.200). Similarly, the homogeneity of the augmentation material was significantly lower in the apical region as compared to the middle (microradiograph *p* < 0.001; histology *p* < 0.001) and cervical parts (microradiograph *p* = 0.040; histology *p* = 0.002). No differences were seen between the cervical and middle regions of the alveoli with respect to homogeneity (Figure 4b).

## 4. Discussion

Questioning the relevance of standardized but unrealistic defects for evaluating biomaterials during preclinical testing [30], the primary goal of this study was to evaluate the homogeneity and density of augmentation materials in simulated ridge preservation procedures. While manufacturers try to optimize the particle size, pore size and interconnectivity of bone substitute materials [16], the handling of the material by clinicians may alter the overall porosity by applying insufficient or excessive compression [20]. Overall, alveolar ridge preservation has been described as an effective therapy preventing bone resorption [3,4], but the use of a particulate synthetic bone substitute has also been shown to interfere with the normal healing processes of alveolar bone [19] and a certain dependency on the exact grafting material used [34] may exist.

Being in line with a clinical report showing that a complete fill of random-shaped defects cannot be achieved predictably [9], less compaction and less homogeneity of augmentation material was seen in the apical parts of the alveoli. The middle third of the alveoli showed the best values for the parameters density and homogeneity, indicating a certain level of predictability. Taking into account inevitable variations in defect sizes following extractions, standard deviations calculated for the volume and mass of augmentation material did not exceed 35%, while ratings for density and homogeneity showed maximum standard deviations of 25%. The weak but mostly significant correlations found among all parameters studied further indicate the reliability of the data presented as well as the comparability of microradiographic and histologic analysis.

A recent animal study has shown that compressive forces in the range of 200 g acting on the crestal surface of an extraction socket are required for a particulate graft material to penetrate into apical areas [24]. While it is generally argued that voids would compromise bone formation, a calcium phosphate cement with an uneven distribution and shape of bubbles performed better in an animal model as compared to more uniform materials [29]. The authors argued that gas bubble formation for the in situ fabrication of optimal porosity would be hardly controllable as gas bubbles move through the augmentation material leading to greater voids in crestal areas [29]. Based on these findings, it may be argued that the overpacking of bone substitutes hindering access to necessary vasculature may be problematic, too [29]. However, a novel mineral–organic osteoconductive adhesive based on tetracalcium phosphate, phosphoserine and water has shown superior regenerative potential without displaying porosity upon placement [35].

Several limitations have to be considered when interpreting the findings presented. The animal model used here differs morphologically from human patients but is in line with a regularly used animal model for preclinical research [30]. As was pointed out in a previous report [36], the socket volume of human teeth is in the range of 0.5 mL, which is greater as compared to this animal model where, in addition, only single roots of teeth have been considered. Limited access to augmentation sites also plays a role in achieving uniform results, which has not been restricted in this experiment. In addition, working on cadaver bone excludes blood flow, which may be problematic in clinical settings. Also, the bone substitute material was not rehydrated prior to use in order to avoid an uncontrollable variable. As such, the results presented may be seen as best-case scenarios obtained under simplified conditions. Furthermore, conducting this study as a live animal experiment would have allowed us to evaluate the relevance of the void spaces seen with respect to the bone response. While the study at hand was aimed at evaluating the extent of graft compression during augmentation procedures, the limitations of the ex vivo study design hinder the transferal of the results into clinical practice immediately. A potential solution for the problem presented here, i.e., non uniformity of augmentation material, may be the in situ formation of a bone substitute, as previously tried for calcium phosphate-based materials [29]. From a methodologic point of view, the results are limited to this specific bone substitute as the shape of the granules may have an effect on defect filling [17]. As shown in a previous study [22], the material used here shows regenerative performance comparable to a more frequently used particulate bovine material, BioOss. Despite maximum care during processing, artefacts resulting from augmentation material being lost during processing cannot be excluded.

## Figures and Tables

**Figure 1 materials-17-00313-f001:**
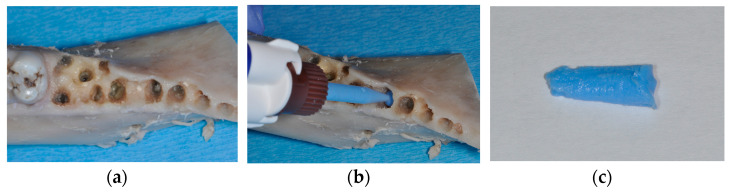
Hemi-mandible of an Aachen minipig with premolars and first molar extracted (**a**). Injection of light body silicone into the alveoli for determining the volume available for augmentation (**b**). Impression material harvested from an alveolus, which was weighed for determining alveolar volume (**c**).

**Figure 2 materials-17-00313-f002:**
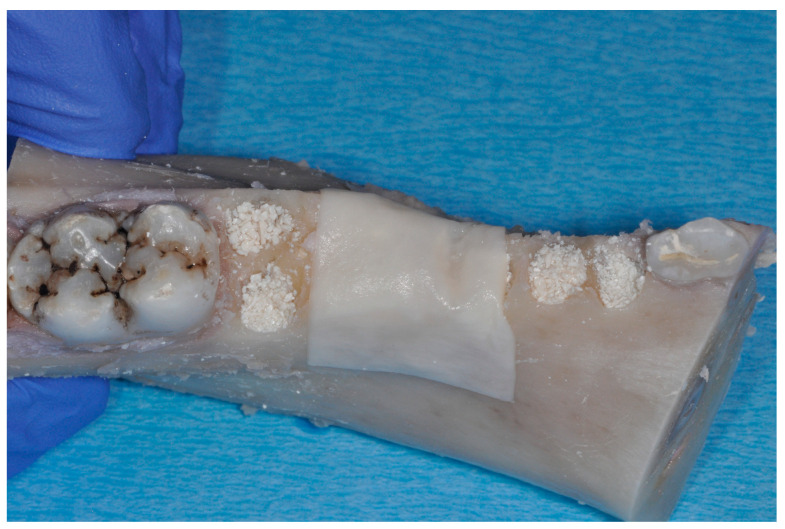
Particulate bone substitute material was used for filling the single alveoli and the wax paper was adapted and secured in order to avoid particles from falling out during further processing.

**Figure 3 materials-17-00313-f003:**
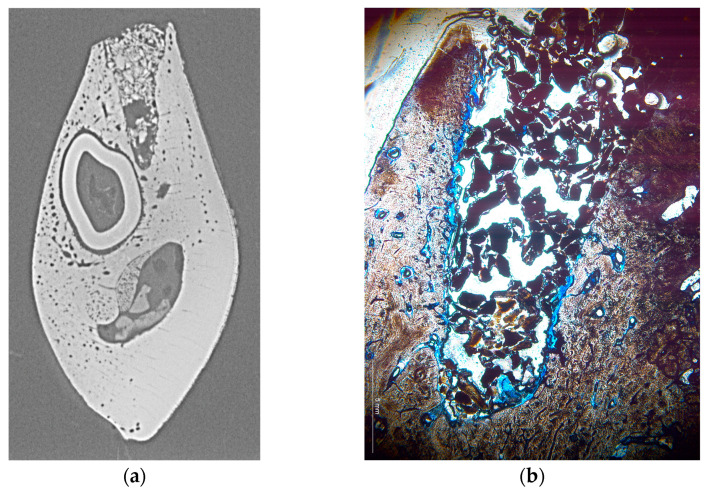
Representative cross-section of two different alveoli shown as microradiograph (**a**) and histologic section (**b**). The microradiograph shows a considerable void within the bone substitute material and high density of the substitute material in the cervical area, while a comparably low density is visible in the histologic section which, however, appears as homogeneous throughout the alveolus.

**Figure 4 materials-17-00313-f004:**
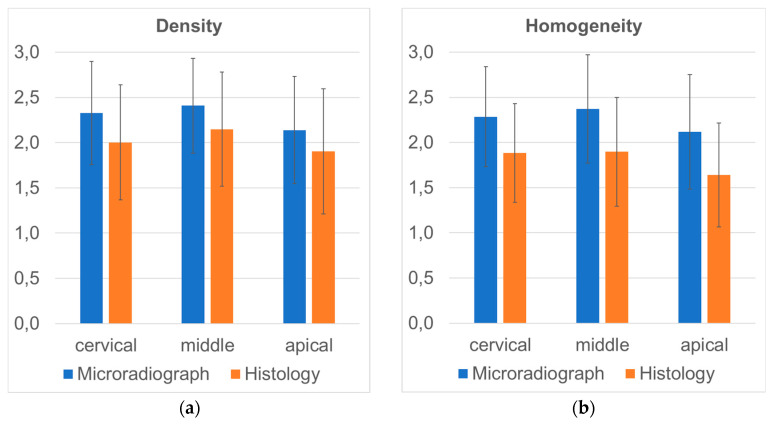
Mean values for the parameter density (**a**) and homogeneity (**b**) of augmentation material recorded in the apical, middle and cervical thirds of alveoli using microradiographs and histology. Note: relative scale used here representing the rating system from 1 to 3.

**Table 1 materials-17-00313-t001:** Mean values and standard deviations for all measurements and ratings.

Parameter	Mean	SD
Volume [cm^3^]	0.120	0.041
Mass Creos [g]	0.155	0.054
Density—microradiograph	2.291	0.395
Density—histology	2.018	0.504
Homogeneity—microradiograph	2.258	0.445
Homogeneity—histology	1.807	0.448

**Table 2 materials-17-00313-t002:** Correlation coefficients (Spearman’s rho) for all parameters evaluated.

	Mass Creos	Density— Microradiograph	Density— Histology	Homogeneity— Microradiograph	Homogeneity— Histology
Volume	0.918	0.995	0.290	0.987	0.227
Mass Creos		0.913	0.262	0.959	0.263
Density— microradiograph			0.279	0.323	0.230
Density— histology				0.267	0.449
Homogeneity— microradiograph					0.233

**Table 3 materials-17-00313-t003:** Mean values, SD and comparisons of apical, middle and cervical thirds of alveoli with respect to density and homogeneity.

	Apical		Middle		Cervical		One Sample Wilcoxon Tests (Corrected *p*-Values)		
	MEAN	SD	MEAN	SD	MEAN	SD	apical—middle	apical—cervical	cervical—middle
Density—microradiograph	2.139	0.591	2.408	0.526	2.327	0.570	<0.001 *	0.100	0.200
Density—histology	1.903	0.692	2.149	0.629	2.003	0.637	<0.001 *	0.300	0.020 *
Homogeneity—microradiograph	2.117	0.634	2.372	0.600	2.285	0.553	<0.001 *	0.040 *	0.200
Homogeneity—histology	1.641	0.575	1.896	0.603	1.883	0.548	<0.001 *	0.002 *	1.000

Significant differences are marked with *.

## Data Availability

Original data are available from the corresponding author upon reasonable request.

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
