# Peer review of "In Vitro Handling Characteristics of a Particulate Bone Substitute for Ridge Preservation Procedures"

_materials, 2024, doi:10.3390/ma17020313_

Round 1

Reviewer 1 Report

Comments and Suggestions for Authors

In this research article authors have discussed about the density and homogeneity of a particulate bone substitute in socket preservation procedures. This research has average novelty and less significant as it is only in vitro study. Several factors need to optimize before going to clinical practice. My comments are as follows: 

  1. 1. Author should organize the figure and put them in result section not in materials and method section.  

  1. 2. What will be critical factors to translate this study into clinical application?

  1. 3. Author should label the Y axis in figure 4.

Comments on the Quality of English Language

The manuscript can accepted with minor revision. 

Author Response

  1. Author should organize the figure and put them in result section not in materials and method section.

We have changed that through revision

  1. What will be critical factors to translate this study into clinical application?

As already mentioned in the discussion section, the findings of this study cannot immediately be transferred into clinical practice. We have added a sentence to this section, which reads: A potential solution for the problem presented here i.e. non uniformity of augmentation material may be the in situ formation of bone substitute as previously tried for calcium phosphate based materials [29]

  1. Author should label the Y axis in figure 4.

Figure 4 is based on the rating system established here and bears no units - we have added this in the description of the figure

Reviewer 2 Report

Comments and Suggestions for Authors

- Please change the title, and throughout the study, to the correct term: "ridge preservation". "Socket preservation" is a wrong term. You are preserving a ridge, not a socket. 

- Correct the "statistical analysis" to "statistical analyses" since you performed more than one tests. 

- Why have you used this cadaver model for your study and not a human cadaver? 

- Have you hydrated the particulate bone graft before you applied it, into the extraction sockets? 

- You need to add the following paper from Thousand et al. 2017, A root volume study of the adult dentition for ridge preservation purposes, General Dentistry, 2017, 65 (5): 21-23to your discussion section and compare it with yours. 

Author Response

- Please change the title, and throughout the study, to the correct term: "ridge preservation". "Socket preservation" is a wrong term. You are preserving a ridge, not a socket. 

Thank you very much for this clarification - we changed throughout the paper

- Correct the "statistical analysis" to "statistical analyses" since you performed more than one tests. 

Changed as suggested

- Why have you used this cadaver model for your study and not a human cadaver?

We have added that this model is in line with a standard animal model used for preclinical testing of bone substitutes; The section under Discussion now reads: “The animal model used here differs morphologically from human patients but is in line with a regularly used animal model for preclinical research [30]”

- Have you hydrated the particulate bone graft before you applied it, into the extraction sockets? 

No, as this would have added an uncontrollable variable - The section under discussion now reads: Also, the bone substitute material was not rehydrated prior to use in order to avoid an uncontrollable variable

- You need to add the following paper from Thousand et al. 2017, A root volume study of the adult dentition for ridge preservation purposes, General Dentistry, 2017, 65 (5): 21-23, to your discussion section and compare it with yours. 

Thanks for pointing us to this nice study which we have added as new reference #36 and discussed under limitations.